# GM1 as Adjuvant of Innovative Therapies for Cystic Fibrosis Disease

**DOI:** 10.3390/ijms21124486

**Published:** 2020-06-24

**Authors:** Giulia Mancini, Nicoletta Loberto, Debora Olioso, Maria Cristina Dechecchi, Giulio Cabrini, Laura Mauri, Rosaria Bassi, Domitilla Schiumarini, Elena Chiricozzi, Giuseppe Lippi, Emanuela Pesce, Sandro Sonnino, Nicoletta Pedemonte, Anna Tamanini, Massimo Aureli

**Affiliations:** 1Department of Medical Biotechnology and Translational Medicine, University of Milano, LITA, Via Fratelli Cervi 93, 20090 Segrate, Milano, Italy; giulia.mancini1681@gmail.com (G.M.); nicoletta.loberto@unimi.it (N.L.); laura.mauri@unimi.it (L.M.); rosaria.bassi@unimi.it (R.B.); domitilla.schiumarini@gmail.com (D.S.); elena.chiricozzi@unimi.it (E.C.); sandro.sonnino@unimi.it (S.S.); 2Section of Clinical Biochemistry, Department of Neurosciences, Biomedicine and Movement, University of Verona, 37134 Verona, Italy; debora.olioso@univr.it (D.O.); mcristina.dechecchi@gmail.com (M.C.D.); giulio.cabrini@univr.it (G.C.); giuseppe.lippi@univr.it (G.L.); 3Section of Molecular Pathology, Department of Pathology and Diagnostics, University Hospital of Verona, 37126 Verona, Italy; 4U.O.C. Genetica Medica, IRCCS Giannina Gaslini, 16147 Genova, Italy; emanuela.pesce@yahoo.it (E.P.); nicoletta.pedemonte@unige.it (N.P.)

**Keywords:** ganglioside GM1, membrane domain, cystic fibrosis, CFTR, correctors, potentiators

## Abstract

Cystic Fibrosis Transmembrane Conductance Regulator (CFTR) protein is expressed at the apical plasma membrane (PM) of different epithelial cells. The most common mutation responsible for the onset of cystic fibrosis (CF), F508del, inhibits the biosynthesis and transport of the protein at PM, and also presents gating and stability defects of the membrane anion channel upon its rescue by the use of correctors and potentiators. This prompted a multiple drug strategy for F508delCFTR aimed simultaneously at its rescue, functional potentiation and PM stabilization. Since ganglioside GM1 is involved in the functional stabilization of transmembrane proteins, we investigated its role as an adjuvant to increase the effectiveness of CFTR modulators. According to our results, we found that GM1 resides in the same PM microenvironment as CFTR. In CF cells, the expression of the mutated channel is accompanied by a decrease in the PM GM1 content. Interestingly, by the exogenous administration of GM1, it becomes a component of the PM, reducing the destabilizing effect of the potentiator VX-770 on rescued CFTR protein expression/function and improving its stabilization. This evidence could represent a starting point for developing innovative therapeutic strategies based on the co-administration of GM1, correctors and potentiators, with the aim of improving F508del CFTR function.

## 1. Introduction

Cystic Fibrosis (CF) is a genetic autosomal recessive disease caused by mutations of the CF transmembrane conductance regulator (CFTR) gene, which leads to impaired ion transport in the epithelial cells of several organs including lung, pancreas, liver and gut. In the lung, this dysfunction induces dehydration of the airway surface, impairing mucociliary clearance. As a result of this condition, mucus accumulates on the bronchial surface, promoting airway obstruction, recurrent bacterial infections, inflammation, and progressive impairment of lung function that ultimately results in limited life expectancy [1,2,3,4]. For the therapy of CF, one of the most promising strategies is the development of small molecules which are able to drive the mutated protein to the plasma membrane (correctors) and increase the open probability of the channel (potentiators) [5,6,7,8,9,10]. One of these, the potentiator VX-770 (Ivacaftor, Kalydeco^®^), is effective at increasing the activity of CFTR protein-bearing gating defects with excellent clinical effects [5,11,12]. Gating mutations are carried by 5% of CF patients. The most common CFTR mutation, F508del, is carried by 70–90% of people with CF, and causes defective biosynthesis, trafficking and stability of the protein [13,14]. To rescue this defect, molecular drugs called correctors (e.g., VX-809 or Lumacaftor and VX-661 or Tezacaftor) have been produced. In addition, F508del CFTR also displays a reduced channel open probability compared with wild-type CFTR [15]. It is now widely acknowledged that the rescue of F508delCFTR requires a combination of correctors and the potentiator VX-770 [16,17,18]. Ivacaftor/Lumacaftor combination therapy (Orkambi^®^) or Ivacaftor/Tezacaftor combination therapy (Symdeko^®^) are available for the treatment of patients who are homozygous for F508del mutation, but the clinical benefits from these treatments are unclear [19,20]. The efficacy of rescue of F508del CFTR by novel correctors has been previously shown to be reduced by CFTR potentiators (e.g., VX-770) [21,22]. Moreover, it has also been described that VX-809 has limited efficacy in stabilizing the rescued F508del CFTR at the cell plasma membrane (PM) [23]. Recently, new generation correctors such as VX-445 (Elexacaftor) or VX-659 were shown to improve F508del CFTR protein processing and trafficking. These drugs, in combination with VX-661 (Tezacaftor) and VX-770 (Ivacaftor), led to increased chloride transport in CF cells [8,10,24]. Thus, major expression of mutated CFTR protein can be attained by using two correctors with different mechanisms of action, although the efficacy was only found to be optimal when the best potentiator so far known, VX-770 (Ivacaftor), was added. This triple combination, called Trikafta^®^, was approved by Food and Drug Administration (FDA) in October 2019, and is also effective in CF patients who harbor only one copy of F508del CFTR [25]. Being a transmembrane protein, the CFTR PM stability and its regulation are the result of a fine coordination of a molecular complex composed of lipids and proteins. 

In airway epithelial cells, CFTR PM stability and function depend on the organization of a multiprotein complex involving F-actin and the scaffolding proteins NHERF1 and ezrin [26,27,28]. This protein complex, besides stabilizing CFTR in highly specialized membrane domains called lipids rafts [29,30], plays an important role in the control of CFTR function, since ezrin, an A- kinase anchoring protein, tethers PKA in the proximity of CFTR, thus allowing cAMP-dependent control of chloride efflux [31,32,33]. Interestingly, the cellular localization of both NHERF1 and phosphorylated ezrin differs between airway cells expressing wt CFTR and F508del CFTR, such that both proteins are mainly localized in the apical region in cells with wt CFTR, whilst they are diffusively present in the cytosol of cells expressing the mutated CFTR [34].

Accumulating evidence suggests that membrane lipids are directly involved in the complex mechanism regulating the NHERF1–CFTR interaction. Indeed, the PDZ domain of NHERF-1 specifically binds cholesterol by the Cholesterol- Recognition-Amino Acid-Consensus (CRAC) motif. Furthermore, the disruption of cholesterol binding activity of NHERF-1 abolishes its colocalization with CFTR protein and reduces channel activity [35]. As recently observed, the potentiator VX-770 is a lipophilic compound that interacts with the lipid bilayer and may bind CFTR lipid interface, probably causing a destabilization of CF membrane that could lead to different organizations of lipids in comparison with normal respiratory cells [36,37,38]. Interestingly, the existence of a direct correlation was recently emphasized between PM levels of the monosialo-ganglioside GM1 and CFTR expression. 

GM1 is a glycosphingolipid formed by a hydrophobic portion called ceramide, and by a hydrophilic head composed of a sequence of five saccharides: glucose, galactose, n-acetyl-neuraminic acid, n-acetyl-galactosamine, and galactose. GM1 is particularly abundant in neurons where it both exerts important physiological properties, ranging from neuronal plasticity to neuroprotection, and modulates the activity of TrkA receptors by direct interaction [39].

CFTR-deficient cells show lower GM1 level compared to normal cells. Moreover, in CFTR- silenced human airway cells, a 60% decrease in GM1 and a parallel reduction of β1-integrin activation and phosphorylation levels of focal adhesion kinase (pFAK) and Crk-associated substrate (pCAS) were observed, partially rescued by the administration of exogenous GM1 [40]. Taken together, these data demonstrate that several factors may be involved in the regulation of CFTR membrane stability by the organization of multiprotein/sphingolipid complexes. 

Therefore, the aim of this study was to explore the effect of GM1 on rescued F508del CFTR protein expression and its activity upon treatment of CF bronchial epithelial (CFBE) cells with CFTR modulators. We found that: (i) GM1 resides in the same PM compartment of WT or rescued F508delCFTR; (ii) GM1 administration to cells, chronically treated with VX-809 and VX-770, increases the level of the mature form of F508delCFTR and scaffolding proteins NHERF1 and p-Ezrin; and (iii) increased level of the mature form of F508delCFTR induced by GM1 is accompanied by an augmented chloride transport. In addition, we demonstrated that in primary bronchial epithelia derived from CF patients, cotreatment with GM1 contrasts the negative effect of VX-770 and restores VX-809 functional rescue.

## 2. Results

### 2.1. The Expression of F508del-CFTR Induces Alteration of the Lipid Composition in Cystic Fibrosis Bronchial Epithelial Cells

Recent literature data shows that the reduced content of CFTR at PM in CFTR-silenced cells is followed by a reduced amount of ganglioside GM1 [40]. In CF, F508del mutation prevents the normal maturation of CFTR protein, which is rapidly degraded, and therefore, is unable to reach the PM. Since this condition resembles that obtained by silencing experiments, we investigated GM1 content in a CF cellular model represented by the CFBE41o^−^ cell line overexpressing F508del-CFTR (F508del-CFBE), and in human primary bronchial epithelial cells derived from patients which were homozygous for F508del mutation (HBE), differentiated at the air–liquid interface. As controls, we used cells expressing the WT form of CFTR. To verify GM1 content in our cell model, we performed immunostaining with cholera toxin that is already known to detect GM1. As revealed by the TLC, endogenous GM1 is characterized by different ceramide moiety due to the presence of very long acyl-chain (upper band) and stearic acid (lower band). Interestingly, as shown in Figure 1a, we found a decreased content of GM1 in F508del-CFBE cells, i.e., by about 40%, compared to the WT-CFBE cell line. Interestingly, a more important reduction was observed in differentiated HBE cells, characterized by decreased GM1 content, i.e., of nearly 80%, in pathological cells compared to WT (Figure 1b). 

Taken together, these results suggest that the expression of F508del-CFTR in bronchial epithelial cells may impair the levels of ganglioside GM1. In addition, the massive reduction observed in HBE, cells committed to form a bronchial epithelium led us to speculate that this ganglioside may play an important role in the homeostasis of this tissue. 

### 2.2. GM1 and WT-CFTR Reside in the Same PM Microdomain

To further investigate the relationship between CFTR and GM1, with particular regard to the possible localization of GM1 in the CFTR lipidic environment, and to verify direct interaction between GM1 and the channel, photolabeling experiments were performed using a GM1 derivative tritium labelled on the sphingosine moiety and carrying a photoactivable group at the end of the fatty acid chain (Figure 2a). The photoactivable group is an azide linked to a nitrophenyl moiety. In this configuration, the azide group is very sensitive to ultraviolet (UV) light. Illumination at λ = 360 nm converts the azide into an unstable nitrene group which can covalently bind the neighboring molecules, including proteins, making them radioactive and consequently detectable by digital-autoradiography (Figure 2b). 

For this purpose, in dark conditions (i.e., under red safe light), ^3^H-GM1-N_3_ was administered to WT- and F508del- CFBE cells. Previous experiments established that the administered GM1 derivative is taken up by the cells, becoming a membrane component diluted into the natural endogenous GM1 ganglioside. After incubation, cells were irradiated under UV and harvested, and proteins were separated by SDS-PAGE. As shown in Figure 2c, in WT-CFBE cells, both band B and band C of CFTR, corresponding to the immature and mature-glycosylated form of CFTR, were detected. Conversely, only the immature form of CFTR was found in F508del-CFBE cells (band B). The same PVDF membrane was subsequently analyzed for the presence of radioactive proteins generated by crosslinking with photoactivable and radioactive GM1. For this purpose, the PVDF membrane was subjected to digital autoradiography using the ^T^Racer digital autoradiograph. As shown in Figure 2c, in WT-CFBE, cells a radioactive band at the same molecular weight of the mature form of CFTR was observed. However, a radioactive signal corresponding to band C of CFTR was almost undetectable in F508del-CFBE cells. These data are consistent with the colocalization of the mature form of CFTR and ganglioside GM1 within the same PM microenvironment.

### 2.3. Chronic Treatment with Potentiator VX-770 Negatively Regulates CFTR Interactome in Bronchial Epithelial Cells Expressing F508del-CFTR

The most promising therapeutic strategies for the treatment of CF encompass restoring the function of mutated CFTR at the apical membrane of epithelial cells. 

A collection of small molecules (correctors) capable of increasing the delivery of mutated CFTR to PM are currently available. Nevertheless, this approach is insufficient to rescue the protein function in cases of the most common mutation, F508del, since the open probability of a mutated channel remains low [21]. 

The most efficacious drug for the activation of CFTR at the PM seems to be the potentiator VX-770 Ivacaftor (Kalydeco^®^, Vertex Pharmaceuticals). The drug increases chloride transport by potentiating the channel-open probability (or gating) with high efficiency for gating mutations such as G551D-CFTR. 

Unfortunately, the combination of VX-770 with the corrector VX-809 showed only a limited effect for F508del mutation, since chronic treatment results in the loss of the beneficial effect, suggesting instability of the channel at cell surface. 

As shown in Figure 3a, even if VX-809 is capable of rescuing, at least in part, F508del-CFTR in F508del-CFBE cells, the combined treatment with VX-770 abolished the effect of VX-809, thus reducing the mature form of rescued F508del-CFTR (band C) at the basal levels. 

The side effect of VX-770 on the destabilization of mutated CFTR at the PM level seems to be due to its interaction with the lipid environment of CFTR [37]. 

We wondered whether VX-770 would also have an impact on GM1 content in CF cells. As shown by immunostaining with cholera toxin, a decrease in GM1 content of nearly 25–30% was observed in F508del-CFBE cells treated with corrector and potentiator individually or in combination (Figure 3b). 

After that, the effect of different treatments on the protein levels of the main CFTR-scaffolding proteins, such as ezrin and NHERF-1, has been analyzed. Western blot analyses revealed that the treatment of F508del-CFBE cells with corrector VX-809 increased the protein level of NHERF-1, t-Ezrin compared to untreated cells. In contrast, the administration of potentiator VX-770 significantly decreased the cellular levels of the scaffolding proteins, a condition preserved also in the presence of combined treatment (Figure 3c). Taken together, these data confirm the negative effect of VX-770 on the mature form of F508del-CFTR rescued using the corrector. In addition, the effect of the potentiator on ganglioside GM1 levels and on CFTR scaffolding proteins led us to speculate that it may interfere with the formation of a proper PM microenvironment, which is essential for the stability and function of the channel.

### 2.4. The Exogenous Administration of GM1 Antagonises the Negative Effect of Potentiator VX-770 on F508del-CFTR Plasma Membrane Stability

To support the active role of GM1 in the CFTR interactome, dose response experiments based on the exogenous administration of ganglioside to F508del-CFBE cells, using a culture medium containing VX-809 and VX-770, were performed. 

The results shown in Figure 4a indicate that treatment with GM1 positively influenced F508del-CFTR maturation. For all the doses used, an increased band C of CFTR was observed compared with untreated cells. Among the different concentrations tested, 50 μM GM1 showed the highest recovery without onset of cell toxicity (Figure 4a,b). For this reason, the concentration of 50 μM was used in subsequent experiments.

First, it was verified that the exogenously-administered GM1 at 50 μM became a component of the cell membrane and rescued the ganglioside deficiency, also upon treatment of cells with VX-809 and VX-770 (Figure 4c). In particular, after ganglioside administration, the levels of GM1 resembled those found in WT-CFBE cells. 

It is worth noting that the administered GM1 is characterized by a main content of stearic acid; for this reason, in treated cells, the lower band results increased.

After that, the previously described photolabeling experiments were performed in F508del-CFBE treated with GM1 in combination or not with VX-809 and VX-770. As shown in Figure 4d, photoactivable GM1 was found to interact with F508del-CFTR only upon exogenous administration of the ganglioside, and the radioactive band C was detectable only in cells fed with 50 µM GM1, both in the presence and absence of VX-809 and VX-770. These data indicate that exogenously-administered GM1 not only becomes a component of PM, but also that it localizes close to the rescued F508del-CFTR as an endogenous counterpart in WT-CFBE cells.

Subsequently, we analyzed the effect of GM1 administration on the levels of CFTR scaffolding proteins NHERF-1, t-Ezrin, and its bioactive form p-Ezrin. As shown in Figure 4e, the negative effect of VX-770 on the scaffolding proteins (Figure 3c) was abolished in the presence of GM1. Increased protein levels of NHERF-1, t-Ezrin and p-Ezrin were also observed in all the conditions tested.

To further support the role of GM1 in the stabilization of the mature form of F508del-CFTR upon rescue with a corrector and potentiator, experiments in the presence of the protein synthesis inhibitor cycloheximide (CHX) were performed. In particular, F508del-CFBE cells were treated with ±VX-809 ±VX-770 ±GM1 for 40 h. After that, CHX was added, and cells were collected at different time points: after 7 h (T_0_), followed by T_1_ at 9 h, T_2_ at 11 h, and T_3_ at 13 h. The mature form of CFTR at different time points was evaluated by western blot analysis. As shown in Figure 4f, GM1 administration reduced the negative effect of VX-770 on the PM stability of F508del-CFTR, corrected by the use of VX-809 at all of the analyzed time points. 

Taken together, these data demonstrate that the exogenous administration of GM1 is effective to rescue the correct chemical physical properties of PM, with the generation of a macromolecular complex that is capable of stabilizing F508del-CFTR, corrected by VX-809, also in the presence of a potentiator.

### 2.5. GM1 Ameliorates the Effectiveness of CFTR Modulators on F508del-CFTR Function

It has been previously shown that the combination of VX-809 and VX-770 in patients carrying F508del-CFTR has limited clinical efficacy [19,20]. Furthermore, it has been demonstrated that chronic treatment with potentiator VX-770 negatively regulates the expression and function of the CFTR protein [21,22]. In this study, we demonstrated that GM1 is reduced in CF cells (Figure 1), and that it resides in the same PM microenvironment as CFTR (Figure 2 and Figure 4). We also observed that GM1 restores the expression and stability of F508del-CFTR upon treatment with VX-809 and VX-770, and increases the expression of NHERF-1 and p-Ezrin involved in the organization of multiprotein complexes which anchor CFTR protein to the cytoskeleton actin of the cell (Figure 4). We hence explored the role of GM1 on CFTR activity upon treatment of CF cells with CFTR modulators. The analysis of CFTR-mediated iodide transport in F508del-YFP CFTR CFBE cells treated with VX-809 or in combination with VX-809 and VX-770 was first carried out, and, as expected, increased CFTR activity due to VX-809 was noted (Figure 5a). In agreement with previous data [21,22], the combination of a potentiator and corrector negatively regulated the CFTR-mediated chloride transport, as shown in Figure 5b. Interestingly, the exogenous administration of GM1 restored CFTR activity, reducing the negative effect of VX-770 (Figure 5c,d). These data suggest a significant role of GM1 content on the PM microenvironment for counteracting the negative effect of VX-770 and maintaining the effectiveness of CFTR modulators on CFTR channel activity.

The ability of GM1 to improve the effectiveness of mutant CFTR rescue by VX-809/VX-770 treatment was further assessed by testing a combination of CFTR modulators in the absence or presence of GM1 on well-differentiated primary cultures of human bronchial epithelial cells, using short-circuit current recordings in a Ussing chamber (Figure 6). To this end, we generated bronchial epithelia from an F508delCFTR homozygous patient. Epithelia were incubated for 48 hr under ALI conditions, with a basolateral medium containing the following single or combo treatments: VX-809 (5 µM), VX-809/VX-770 (5 µM/5 µM), VX-809/VX-770/GM1 (5 µM/5 µM/50 µM) or with vehicle alone (DMSO). Epithelia were then mounted in Ussing chambers to measure chloride secretion by short-circuit current analysis (Figure 6a). After blocking the Na^+^ current with amiloride, epithelia treated with DMSO showed small response to CPT-cAMP (a membrane permeable cAMP analogue) and VX-770, or to selective CFTR inh-172 (Figure 6a,b). Treatment with VX-809 for 48 hr resulted in significant rescue of mutant CFTR-mediated chloride current, as seen by the increased current inhibited by CFTR inh-172. Cotreatment with VX-809 plus VX-770 caused a significant decrease of mutant CFTR rescue compared to treatment with VX-809 (Figure 6a,b). However, when VX-809/VX-770 was administered together with GM1, the rescue of mutant activity by corrector treatment was restored (Figure 6a,b). 

## 3. Discussion

Accumulating evidence suggests the active role of sphingolipids in CF disease [41]. In particular, a direct relationship between CFTR and sphingolipids in human bronchial epithelial cells was reported in 2014 by Itokazu et al. The authors demonstrated that CFTR-silenced cells are characterized by a 60% decrease in the glycosphingolipid GM1 content, which is responsible for reduced β_1_-integrin activation with a consequent decreased phosphorylation of FAK and CAS, and the blocking of cell motility [40]. 

It is worth noting that the addition of GM1 to CFTR-silenced cells was able to revert the phenotype [40]. Our studies provide the first in vitro evidence that GM1 could stabilize the mutated CFTR on the PM of cells. We show here that GM1 reduction occurs also in CF bronchial epithelial cells. Indeed, as shown in Figure 1, we found a reduction of GM1 cellular levels in both CFBE41o^−^ and HBE cells expressing F508del CFTR, supporting the direct correlation between ganglioside expression and the presence of the mature form of CFTR at the PM. 

These data and the association of CFTR with lipid rafts led us to speculate that the channel and GM1 reside in the same PM microenvironment [42]. 

To investigate this, we used photolabeling technology using a radioactive and photoactivable GM1 derivative, and demonstrated that in bronchial epithelial cells, CFTR and GM1 reside in the same PM microenvironment, with correlation missing in the case of cells expressing F508del-CFTR (Figure 2).

Since the ganglioside GM1 is an important bioactive lipid for the control and stabilization of several proteins, these results present a new scenario in the study of CFTR, and include the ganglioside in the cluster of molecules belonging to CFTR interactome that change in the presence of F508del mutation.

CFTR plasma membrane stabilization is an important issue, also for the use of the correctors and potentiators. Indeed, the chronic use of VX-770, a component also in the recent formulation Trikafta^®^, does not maintain high levels of the mature form of F508del CFTR on the PM (Figure 3). This unexpected effect is probably due to the fact that VX-770 negatively regulates the levels of the CFTR scaffolding proteins NHERF-1 and Ezrin, as well as GM1 (Figure 3). 

The emerging evidence of a direct role of plasma membrane GM1 in the regulation of CFTR stability and function suggest that its exogenous administration may be a valid approach in this sense. The modulation of the composition of the PM sphingolipid represents an important issue for hydrophobic lipids. In contrast, gangliosides are soluble in cell culture medium, where they form micelles which are capable of exchanging monomers with the PM, and thus increasing cell membrane concentration. Here, we found that exogenous GM1 restores GM1 cellular levels similar to those found in bronchial cells expressing WT CFTR (Figure 4). Importantly, restoring the PM concentration of GM1 ameliorates the stability of F508del-CFTR and of its interactome, also in the presence of a potentiator, with significant improvement in the chloride transport of the rescued mutated channel (Figure 4 and Figure 5). The same effect was also observed in primary bronchial cells (Figure 6).

These data open a new scenario related to the CFTR PM interactome, indicating that only a fine coordination between lipids, such as GM1, and a selected pattern of proteins creates the proper PM environment that is fundamental for CFTR stability and function. 

## 4. Conclusions

Several lines of evidence for CFTR interactome focus on the protein side, whilst there is only a modest amount of information on the lipid environment. CFTR, like other proteins associated with PM, belongs to specific membrane lipid domain called lipid rafts, where only tight interactions with a specific set of proteins and lipids are able to organize the proper macromolecular complex responsible for preserving channel activity. 

Using a radioactive and photoactivable GM1 derivative, we demonstrated that GM1 and CFTR reside in the same PM microenvironment in bronchial epithelial cells, and that segregation is missed in case of cells expressing F508del-CFTR (Figure 2).

Since ganglioside GM1 is an important bioactive lipid involved in the control and stabilization of several proteins, these results open a new scenario in the study of CFTR, including the ganglioside in the cluster of molecules belonging to CFTR interactome which are modified in the presence of the F508del mutation. Both the CFBE41o^−^ cell line and primary bronchial epithelial cells differentiated at the air–liquid interface expressing F508del-CFTR are characterized by low contents of GM1 compared to the WT counterpart, as shown in Figure 1. 

The susceptibility of CFTR to a lipid environment to control its PM stability is also highlighted by the negative effect of potentiator VX-770 on F508del-CFTR rescued by the correctors. Due to its high hydrophobicity, VX-770 accumulates in the lipid core of PM close to CFTR, thus altering the physical and chemical properties of the membrane [39], and impairing the formation of the macromolecular complex involving CFTR-scaffolding proteins such as Ezrin and NHERF-1 (Figure 3). Interestingly, the recovery of GM1 PM levels by its exogenous administration gives rise to the proper organization of the CFTR plasma membrane environment, which stabilizes the rescued F508del-CFTR protein and its interactome, also in the presence of the potentiator; this is associated with significant improvement in the chloride transport of the rescued mutated channel (Figure 4 and Figure 5). It is worth noting that the ability of GM1 to improve mutant CFTR rescue by VX-809/VX-770 combo was confirmed also on primary bronchial epithelia derived from a F508del/F508del CF patient (Figure 6).

These data clearly indicate that CFTR protein needs a specific PM environment to exert its function, and that GM1 seems to play a key role therein. In addition, since VX-770 is also used for the new FDA-approved pharmacological combination Trikafta^®^ [25], study on the adverse effects of a potentiator on CFTR plasma membrane stability remains a high priority, and could be useful to better address new therapeutic strategies for CF. 

## 5. Materials and Methods 

### 5.1. Cell Models

Human bronchial epithelial cells CFBE41o^−^ stably overexpressing WT-CFTR (WT-CFBE) and F508del/F508del-CFTR (F508del-CFBE) were grown in Eagle’s-minimum essential-medium E-MEM, supplemented with 10% of FBS, 2 mM glutamine, Penicillin/Streptomycin (100 u/mL and 100 µg/mL respectively) and 2 μg/mL puromicyn for F508del-CFBE and 0.5 μg/mL for WT-CFBE. 

CF human bronchial epithelial cell line CFBE41o- stably coexpressing human F508del-CFTR (CFBE-F508del) and the high-sensitivity halide-sensing green fluorescent analog yellow fluorescent protein (HS-YFP) YFP-H148Q/ I152L were grown on substrates coated with an extracellular matrix containing fibronectin/bovine collagen type I/BSA in MEM supplemented with 10% fetal calf serum, 2 mM L-glutamine, Nonessential Aminoacid 100x (Euroclone) and 2 µg/mL of Puromycin and 750 µg/mL of Geneticin (G418) as positive selection of expression of F508del CFTR and YFP, respectively [43]. 

Human primary bronchial epithelial cells (HBE) derived from healthy subjects (WT) and CF patients who were homozygous for F508del mutation (F508del) were obtained from “Servizio Colture Primarie” of the Italian Cystic Fibrosis Research Foundation. These cells were grown as described elsewhere [44]. After a proliferative stage, cells were plated in transwell supports (500,000 cells for cm^2^) and induced to differentiate by using a differentiating medium. Cells at air-liquid interface were used after 14–16 days from the transwell seeding, once complete differentiation had occurred.

### 5.2. Cell Treatment with CFTR-Modulators

Ivacaftor (VX-770) and Lumacaftor (VX-809), purchased from Selleckchem, were solubilized in dimethylsulfoxide (DMSO) (Sigma) and administered directly into the cell medium at final concentrations of 5 μM and 1 μM (5 μM for primary cells) respectively. As a control, cells were treated with the same volume of DMSO.

### 5.3. Cell Feeding with Ganglioside GM1

GM1 was solubilized in the complete cell culture medium at final concentrations of 10, 50 and 100 µM. In detail, an appropriate amount of GM1 powder was solubilized in methanol in a glass tube and dried under gentle N_2_ flux. After total evaporation of the solvent, complete cell medium was added and 5 cycles of shaking and sonication were performed, thus allowing complete solubilization of the ganglioside to occur [45,46].

### 5.4. MTT Assay

Cells were plated at a density of 13,500 cells/cm^2^ in a 96-well microplate, and subjected to the different treatments for 48 h. At the end of incubation, the medium was removed and cells were washed with PBS. After washing, 71 µL of MTT solution was added to each well, prepared by dissolving thiazolyl blue tetrazolium bromide in PBS at a concentration of 4 mg/mL. They were then dissolved in cell culture medium at a 1:4 ratio. The cells were incubated for 4 h in a humidified atmosphere at 37 °C and 5% CO_2_. After that, MTT solution was removed, and 57 µL of lysis buffer was added (95% isopropanol, 5% formic acid). After shaking for 10 min, the concentration of formazan salts formed was evaluated using a microplate reader Victor (Perkin-Elmer) at λ = 570 nm. 

### 5.5. SDS-PAGE and Western Blotting

The protein content of cell lysates was evaluated through DC protein assay (Biorad) according to manufacturer’s instructions. Aliquots of cell lysates corresponding to the same amount of proteins were resuspended in Laemmli buffer and denaturated for 10 min at 100 °C. The sample dedicated to the analyses of CFTR were resuspended in Laemmli buffer and treated at 40 °C for 10 min. The electrophoresis run was performed using a Miniprotean II unit, produced by Bio-Rad using a gradient gel of 4–20% of poly-acrylamide (Bio-Rad). 

After electrophoresis separation, proteins were transferred to PVDF membrane in 5% skim milk in TBS-T 0.1%. The PVDF was then washed 3 times with TBS-T 0.1% and incubated overnight at 4 °C with the appropriate primary antibodies. CFTR antibody, purchased from the Cystic Fibrosis North American Foundation, was used at final dilution of 1:2000, whilst NHERF-1, t-Ezrin and p-Ezrin antibodies (BD Sciences) were used at final dilution of 1:500. GAPDH antibody (Sigma) was used at dilution of 1:7000 and Calnexin antibody (BD Sciences) at dilution of 1:1000, respectively. The PVDF was finally washed 3 times with TBS-T 0.1% and incubated for 1 h at RT with appropriate secondary antibodies. The membrane was then washed again for 3 times and the peroxidase activity was assessed through incubation with horseradish peroxidase substrate (Westar Cyanagen). The chemiluminescent signal was revealed using a Mini HD9 (UviTec, Cambridge) and analyzed by Nine Alliance mini HD9 software [27,47].

### 5.6. TLC Immunostaining

Lipids from the lyophilized samples were extracted with chloroform:methanol:water 20:10:1 (*v*:*v*:*v*) and subjected to a two-phase Folch’s partitioning to obtain the aqueous (AP) and organic phases (OP). Gangliosides contained in aliquots of AP corresponding to 1 mg of cellular proteins for CFBE41o^−^ and 0.5 mg of cellular proteins for HBE were separated by thin layer chromatography (TLC) using the solvent system chloroform:methanol: 0.2% aqueous CaCl_2_ 50:42:11 (*v*:*v*:*v*). Cholera toxin binding to GM1 was assessed by TLC immunostaining following standard protocols [48]. Briefly, after chromatographic separation, the TLC plates were coated with polyisobuthylmethacrylate solution three times, and air dried for 1 h before being immersed in blocking solution (3% BSA in PBS) for 1 h. The plates were then incubated with cholera toxin at 10 μg/mL in 1% BSA in PBS for 1 h at RT. After incubation, two washes with PBS were performed, and immunoreactive bands were revealed using o-phenylenediamine (OPD)/H_2_O_2_ in 0.05 M citrate-phosphate buffer pH 5.0 [49].

### 5.7. Photolabelling Experiments

[3-^3^H-sph]-GM1-N_3_ was administered at a final concentration of 1.25 µM in the presence of the same amount of normal GM1. Briefly, under dark conditions, the appropriate amount of [3-^3^H-sph]-GM1-N_3_ and purified GM1 dissolved in 2-propanol was transferred into a sterile glass tube and dried under a nitrogen stream. The powder was then solubilized in an appropriate volume of prewarmed (37 °C) medium (E-MEM) and administered to the cells previously plated in petri plates and subjected to different treatments. After 6 h in dark conditions, the medium was collected and its radioactivity was measured. The cells were washed with medium containing FBS for 30 min to remove the photoactivable GM1 remaining nonspecifically attached to the cells, and then with PBS containing Na_3_VO_4_. Finally, cells were illuminated with a UV lamp for 40 min. The cells were then collected in PBS containing Na_3_VO_4_, and lysed to perform SDS-PAGE analysis. The radioactivity associated with the protein was detected using a digital-autoradiograph ^T^Racer (Biospace Lab) [45].

### 5.8. Cycloheximide Treatment

The day after seeding on plates, cells were subjected to the different treatments with corrector (VX-809) and/or potentiator (VX-770) and/or exogenous GM1. Cycloheximide was added after 40 h at a concentration of 100 μg/mL. Cells were collected after different time points, i.e., 7 h (T_0_), 9 h (T_1_), 11 h (T_2_) and 13 h (T_3_), and the CFTR content was evaluated by western blot analysis [47].

### 5.9. CFTR Function Assay

For iodide influx experiments, 15,000 YFP CFBE-F508del cells were seeded on round glass coverslips in the absence of positive selection antibiotics and preincubated for 48 h with vehicle, VX-809 or VX-809, in combination with VX-770, or VX-809 in combination with VX-770 and GM1. At the time of assay, cells were washed in Dulbecco’s PBS (in mM: 137 NaCl, 2.7 KCl, 8.1 Na_2_HPO_4_, 1.5 KH_2_PO_4_, 1 CaCl_2_, and 0.5 MgCl_2_, pH 7.4) and subsequently incubated in 1 mL volume in a Medical Systems perfusion chamber with stimulation cocktail (20 µM forskolin and 5 µM VX 770) in the presence or absence of 10 µM CFTR inhibitor, CFTRInh-172 (Sigma), for 30 min. The time courses were performed at 25 °C and a baseline signal was acquired before adding the stimulus. Fluorescence was measured with a Nikon TMD inverted microscope through a Nikon Fluor 40X objective. The signal was acquired with a Hamamatsu C2400-97 CCD intensified video camera (Hamamatsu City, Japan) at a rate of 1 frame/20 s with an integration time ranging from 0.1 to 1 s. The fluorescence coming from each single cell was analyzed using customized software (Spin, Vicenza, Italy). The YFP fluorescence decay rate was calculated by fitting the fluorescence data of the time courses by means of an exponential function. CFTR activity was calculated as the difference between the YFP fluorescence decay rate in the absence or presence of CFTRInh-172. The assay of each sample consisted of continuous 120-s fluorescence readings with injection, 40 s before and 80 s after, of a modified Dulbecco’s PBS enriched in iodide (in mM: 137 NaI, 2.7 KCl, 8.1 Na_2_HPO_4_, 1.5 KH_2_PO_4_, 1 CaCl_2_, and 0.5 MgCl_2_, pH 7.4), to reach a final iodide concentration of 50 mM, as described [7]. The results are presented as transformed data to obtain the percentage signal variation (*Fx*) relative to the time of addition of the stimulus according to the equation: (1)Fx=[(Ft−Fo)/Fo]×100
where *Ft* and *Fo* are the fluorescence values at time *t* and at the time of addition of the stimulus, respectively.

### 5.10. Short-Circuit Current Recordings

Snapwell inserts carrying differentiated bronchial epithelia were mounted in a vertical Ussing chamber. Both hemichambers were filled with 5 mL of a solution containing (in mM) 126 NaCl, 0.38 KH_2_PO_4_, 2.13 K_2_HPO_4_, 1 MgSO_4_, 1 CaCl_2_, 24 NaHCO_3_ and 10 glucose, and both sides were continuously bubbled with a 5% CO_2_/95% air mixture, while the temperature of the solution was kept at 37 °C. The transepithelial voltage was short-circuited with a voltage-clamp (DVC-1000, World Precision Instruments) connected to the apical and basolateral chambers via Ag/AgCl electrodes and agar bridges (1MKCl in 1% agar). The offset between the voltage electrodes and the fluid resistance was adjusted to compensate for the parameters before carrying out the experiments. The short-circuit current was recorded with a PowerLab 4/25 (ADInstruments) analog-to-digital converter, connected to a Macintosh computer.

Data are expressed as means ± SEM, *n* = 5 independent epithelia (biological replicates) per condition.

### 5.11. Statistical Analysis

All experiments were performed in triplicate and, unless indicated differently in the text and figure legend, each result is the mean of three independent experiments. Statistical significance was assessed by Student-Neumann-Keuls post hoc test (comparison between two groups) and by one-way or two-way ANOVA for more than two groups (followed by Turkey or Dunnett Neueman-Keuls on Bonferroni post hoc test) unless otherwise indicated in the figure legends. A *p* value <0.5 was assumed to be statistically significant.

## Figures and Tables

**Figure 1 ijms-21-04486-f001:**
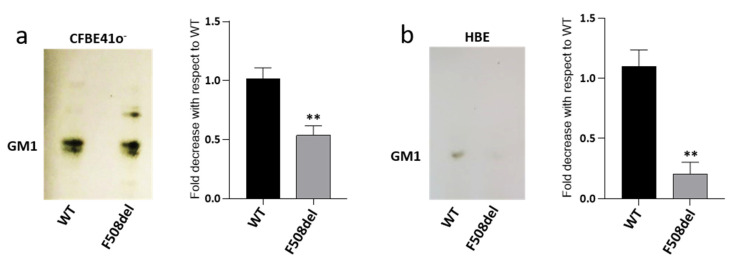
GM1 levels are decreased in both CFBE41o^−^ and human primary bronchial epithelial cells expressing F508del CFTR. GM1 of CFBE41o^−^ overexpressing the WT or F508del form of CFTR, (**a**) and of human primary bronchial epithelial cells (HBE) derived from CF (F508del) and non-CF (WT) subjects; (**b**) Gangliosides were extracted from cells, separated by thin layer chromatography (TLC) and visualized by TLC immunostaining using cholera toxin. GM1 levels were quantified using ImageJ software and represented in the graphs as fold decrease with respect to WT. ** *p* < 0.003 vs. WT.

**Figure 2 ijms-21-04486-f002:**
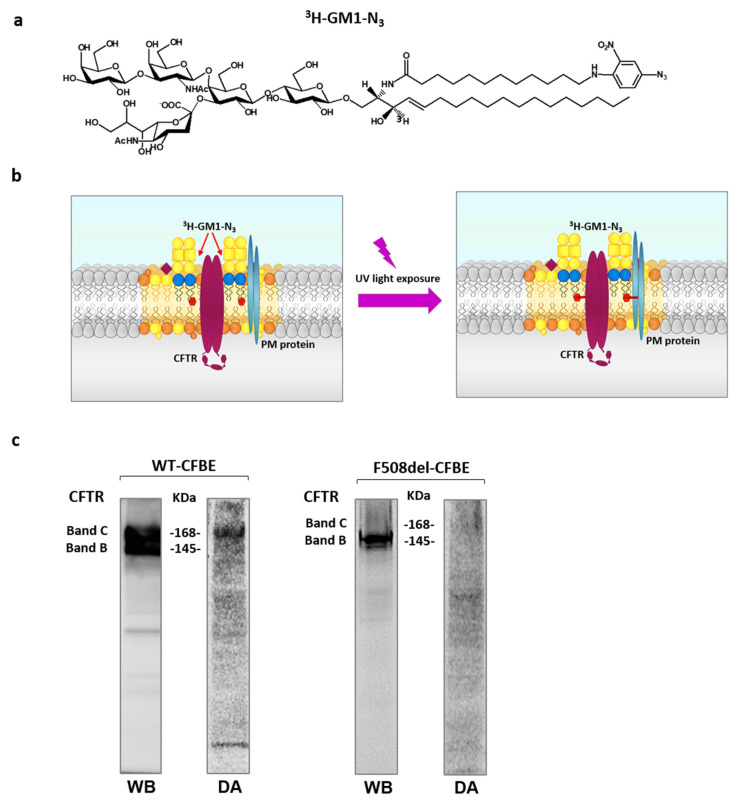
Ganglioside GM1 and CFTR reside in the same PM microenvironment. Chemical structure of the radioactive and photoactivable ganglioside GM1 derivative (^3^H-GM1-N_3_) (**a**) and schematic representation of its use in a photolabeling experiment. (**b**) CFBE41o^−^ cells overexpressing the WT form of CFTR or F508del CFTR were treated with tracer quantity of ^3^H-GM1-N_3_ in dark conditions and, after 6 h, cells were illuminated under UV light, harvested and subjected to SDS-PAGE and immunoblotting analyses against CFTR (WB). The same PVDF membranes were then subjected to digital autoradiography (DA) to reveal the radioactivity associated with the proteins due to the cross link with the ^3^H-GM1-N_3_ (**c**).

**Figure 3 ijms-21-04486-f003:**
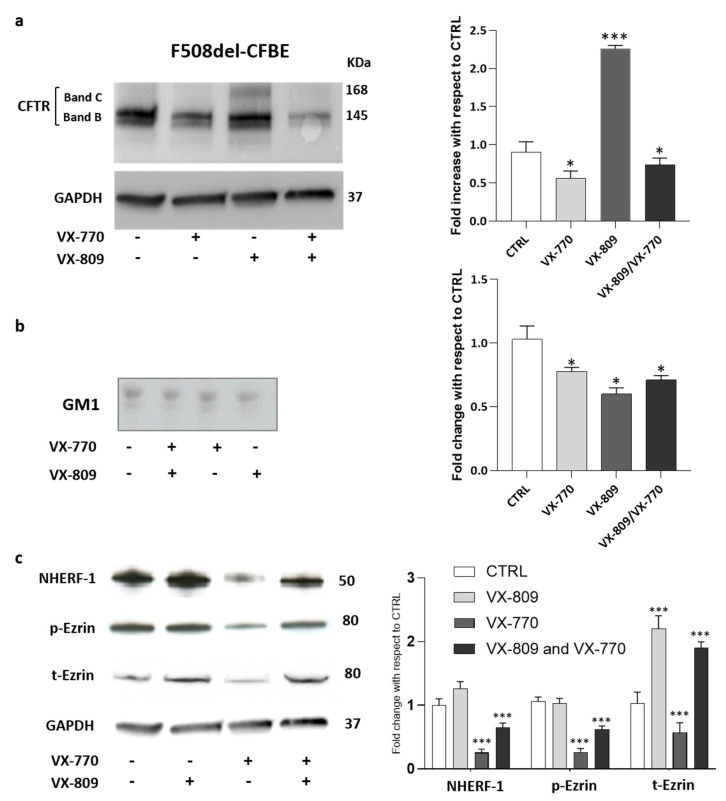
Chronic treatment with VX-770 negatively regulates GM1 levels and the F508del CFTR maturation including the expression of its scaffolding proteins. CFBE41o- cells overexpressing F508del CFTR were treated for 48 h with 5 µM VX-770, 1 µM VX-809 alone or in combination and subjected to the different analyses: (**a**) representative western blot against CFTR and its quantification; (**b**) representative TLC immunostaining with cholera toxin in order to specifically detect GM1 followed by its quantitative graph; and (**c**) representative western blot against CFTR scaffolding proteins and its quantification. * *p* < 0.03 vs. CTRL, *** *p* < 0.0001 vs. CTRL.

**Figure 4 ijms-21-04486-f004:**
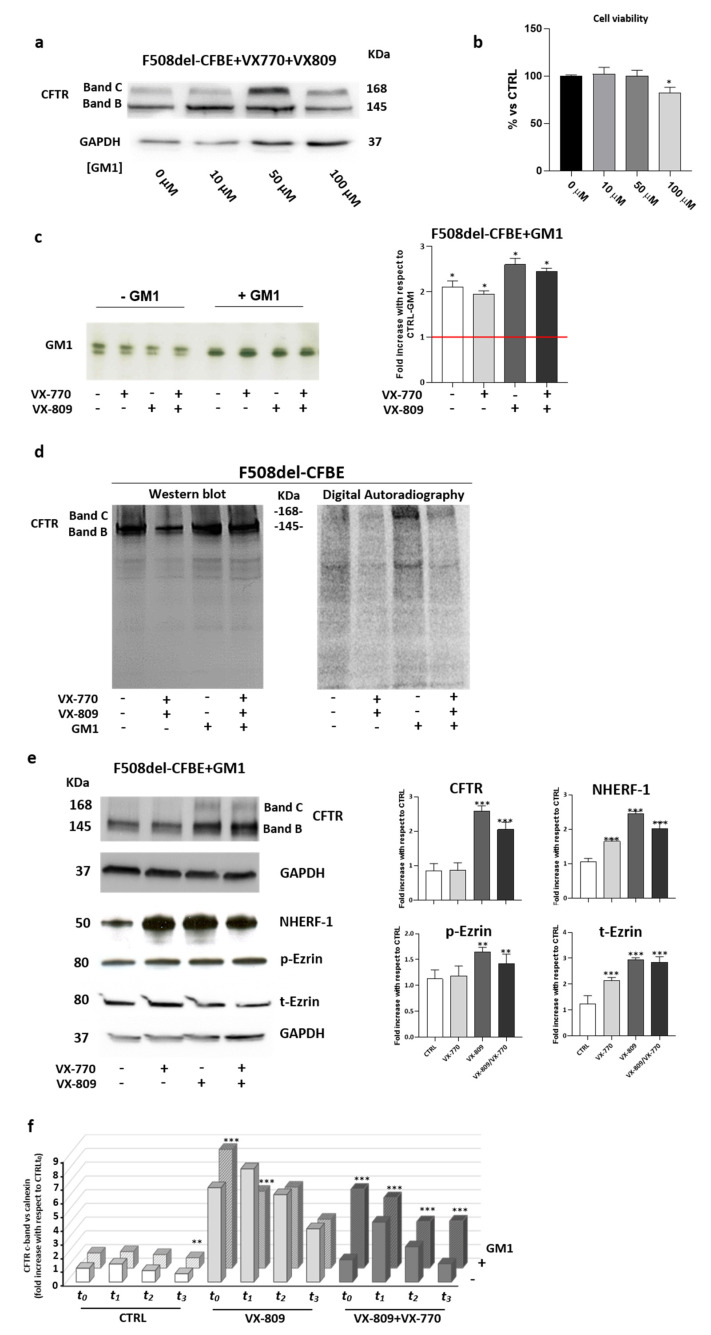
Restoring the GM1 content in bronchial epithelial cells compensates for the adverse effect of VX-770 on the plasma membrane stability of F508del CFTR. (**a**) Representative western-blot images show the expression of F508del CFTR protein in F508del-CFBE cells subjected to treatment for 48 h, with 1 µM corrector (VX-809), 5 µM potentiator (VX-770) and GM1 at concentrations of 10, 50 and 100 μM. GAPDH was evaluated as a loading control. (**b**) Cell viability of F508del-CFBE treated with increasing concentrations of ganglioside GM1. Cells were seeded at a density of 13500 cells/cm^2^ in a 96-well plate, and after 48 h of incubation, cell viability was evaluated by MTT assay. Data are expressed as % vs. untreated cells. * *p* < 0.05 vs. CTRL (0 μM). (**c**) F508del-CFBE cells were treated with VX-809 (1 µM) and VX-770 (5 µM), individually or in combination, and subjected or not to exogenous treatment with 50 µM GM1. Ganglioside GM1 was visualized by TLC immunostaining using cholera toxin. GM1 levels were quantified using the ImageJ software and represented in the graph as fold increases with respect to untreated F508del-CFBE cells * *p* < 0.05 vs. CTRL without GM1. (**d**) F508del-CFBE cells were treated with VX-809 (1 µM) and VX-770 (5 µM) in the presence or absence of 50 μM GM1 for 48 h. Then, the cells were treated with a tracer quantity of ^3^H-GM1-N_3_ in dark conditions, and after 6 h, cells were illuminated under UV light, harvested and subjected to SDS-PAGE and immunoblotting analyses against CFTR. After this, the same PVDF membrane was subjected to digital autoradiography to reveal the radioactivity associated with proteins due to the cross link with the ^3^H-GM1-N_3_. Left: representative immunoblotting analysis of CFTR; right: autoradiography of the same PVDF membrane. (**e**) Representative immunoblotting and densitometric analyses of CFTR, NHERF-1, p-Ezrin and t-Ezrin proteins performed in the whole lysates of F508del-CFBE cells loaded with 50 μM GM1 and subjected or not to the treatment with VX-809 (1 μM) and VX-770 (5 μM), individually or in combination. Protein content was normalized on the loading control GAPDH and compared to the CTRL** *p* < 0.001 vs. CTRL, *** *p* < 0.0002 vs. CTRL. (**f**) Graph reporting the quantification of the western-blot analyses of F508del-CFTR evaluated in F508del-CFBE cells treated with ±VX-809 (1 µM) ±VX-770 (5 µM) ±GM1 (50 µM) and with cycloheximide (CHX, 100 µg/mL) for different time points: T_0_: 7 h, T_1_ 9 h, T_2_ 11 h, and T_3_ 13 h. The data are expressed as fold increases vs. control (CTRL) of the mature form of F508del-CFTR for each time point, normalized on the loading control calnexin (CNX). ** *p* < 0.003 vs. CTRL, *** *p* < 0.0005 vs. CTRL.

**Figure 5 ijms-21-04486-f005:**
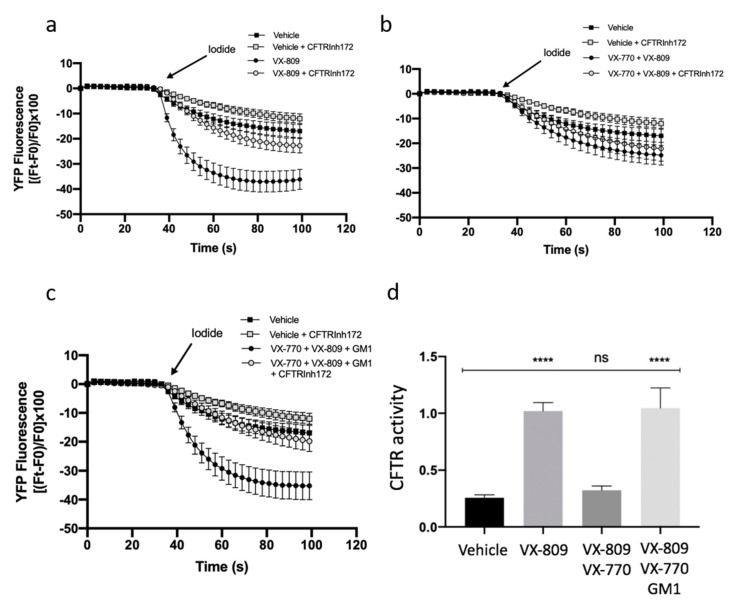
GM1 reduces the negative effect of VX-770 on CFTR-dependent chloride efflux in CFBE41o^−^ -YFP-F508del cells. CFTR-dependent chloride efflux was assayed by single cell fluorescence imaging analysis of YFP fluorescence quenching by iodide, stimulated by forskolin (20 µM), in the presence or absence of CFTRInh-172 (10 µM). Each point in the representative traces is the mean ± SEM of data coming from all cells in the field (5–10 cells) (**a**) Representative traces showing iodide influx in control conditions (vehicle), or after 48 h incubation with 1 µM VX-809; (**b**) Representative traces showing iodide influx in control condition (vehicle), or after 48 h incubation with 5 µM VX-770 in combination with 1 µM VX-809; (**c**) Representative traces showing iodide influx in control condition (vehicle), or after 48 h incubation with 5 µM VX-770 in combination with 1 µM VX-809 and 50 µM GM1. (**d**) The YFP fluorescence decay rate measured in each cell was calculated by fitting data of time courses by an exponential function, both in the absence and presence of CFTRInh-172. CFTR activity was obtained by subtracting the data of cells treated with CFTR inhibitor 172 from those of cells in the absence of the inhibitor. Each bar corresponds to the mean ± SEM of data points coming from at least three different experiments (5–10 different cells/each experiment). Statistical comparisons were made using a nonparametric ANOVA test (**** *p* < 0.0001).

**Figure 6 ijms-21-04486-f006:**
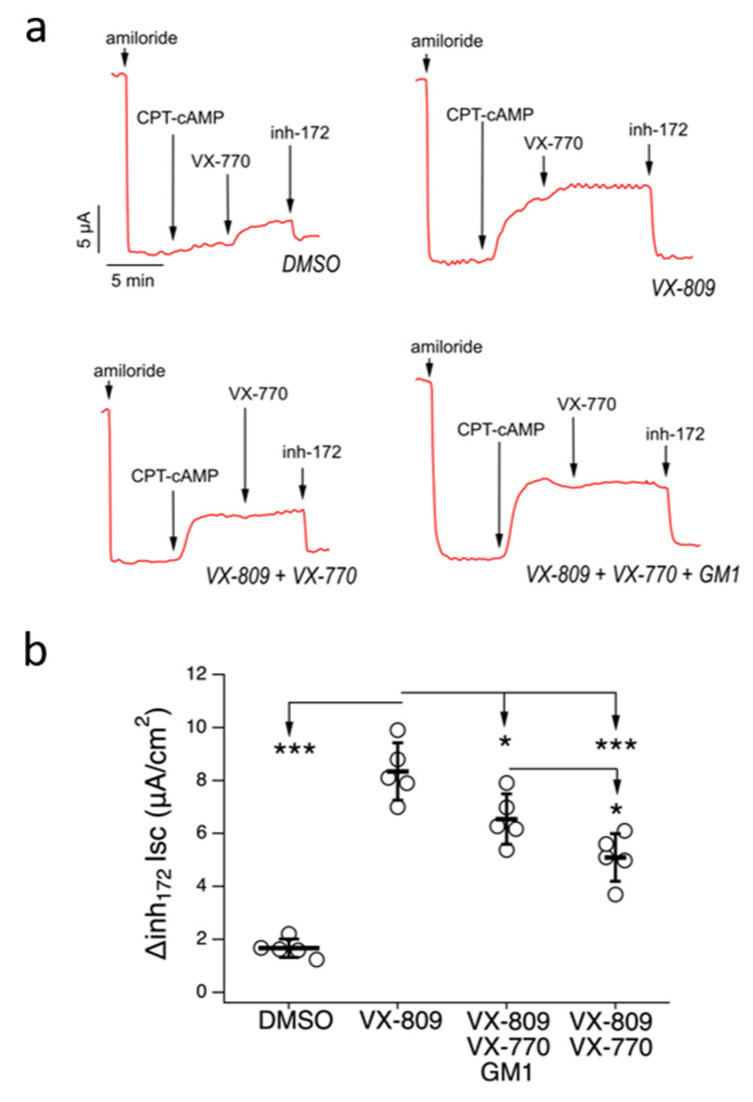
GM1 improves F508del rescue by combo VX-809/VX-770 treatment in primary bronchial epithelia. Representative traces (**a**) and dot plot (**b**) summarizing data from the Ussing chamber recordings of human primary bronchial epithelia from a homozygous F508del patient, treated with VX-809 (5 µM), VX-809/VX-770 (5 µM/5 µM), VX-809/VX-770/GM1 (5 µM/5 µM/50 µM) or DMSO. Symbols indicate statistical significance: *** *p* < 0.001, * *p* < 0.05.

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
