# Peer review of "GM1 as Adjuvant of Innovative Therapies for Cystic Fibrosis Disease"

_ijms, 2020, doi:10.3390/ijms21124486_

Round 1

Reviewer 1 Report

This is an interesting piece of work but many important points need clarifications. The main result is that the ganglioside GM1 could stabilize the plasma membrane form of mutated CFTR. I have listed below my major and minor comments

Major

Western blots fig 3b and 4c are similar. This is not acceptable

No indication of the number of experiments. In the method section it is said triplicate but from how many different experiments?

The Western blot data are not convincing :

  • Why the c band is present in Fig 4a but not fig 3a lane 4. The c band is also comparable Fig 4e lane 3 and 4 in the presence of GM1
  • GM1: two bands are apparent. How do you determine the fold change?

CFTR activity: figure 5d What is this? How do you calculate this parameter from a, b and c? why the effect of CFTRinh172 is similar with vehicle and VX770+VX809 (fig 5a,b and c) but why vehicle+172 and VX770+809+172 are not similar. Does this mean that 172 is not able to block 100% of the rescued CFTR activity? Is there a residual activity of CFTR even in the vehicle condition?

Minor

Introduction: add a description of the ganglioside GM1. For example it is very abundant in neuronal plasma membrane with important physiological functions

Line 90: add Liu et al 2019, the science paper showing binding of vx770

Lines 103 to 106: recovery is not appropriate to describe the effect of GM1

Line 178: add a ref for the CFTR Po

Fig Ussing chamber: add the time scale and Isc (is it corrected by the surface?)

Fig 5d: 4 stars?

Method: Ussing experiments. Please confirm that a symetrical ionic solution is used here?

Reviewer 2 Report

This is a well-written paper focusing on an interesting topic, that is the possible role of ganglioside GM1 as an adjuvant to increase the effectiveness of CFTR modulators. I believe that reported results may arouse interest in the readers of IJMS.

Major comments

I would suggest that the Authors separate the “Results” and “Discussion” section in their manuscript to give more continuity to the discussion.

Reviewer 3 Report

This paper shows the importance of the lipid micro-environment for the rescue of the mutant CFTR. Apparently, ivacaftor destabilises the membrane, reducing the GM1, and consequently limits the rescue efficacy of lumacaftor. Most interesting, feeding the cells with GM1 reverts the negative effect of ivacaftor.

It is an excellent piece of work, very well designed and clearly presented. It should be accepted almost immediately.

Only few minor comments:

Figure 1a: what is the extra band of F508del gangliosides? degraded GM1, or a ganglioside specie absent in WT?

Figure 1b GM1 of F508del cells of HBE is not visible to the eye in the TLC image. A quick densitometric analysis of the image indicates that the GM1 is ~11% for the mutant. Perhaps a more representative figure should be showed.

Figure 1, 3, 4: number of experiments should be indicated. Also the meaning of the error bars should be indicated (standard deviation or standard error?).

Figure 3a: A quantification should be desirable.

Explanation of the two GM1 bands is in section 2.4. Perhaps it should be anticipated in a previous section, as double bands are present from figure 1 in section 2.1.

Figure 4: MTT is a test that evaluates the mitochondrial activity, normally related with the cell viability. However, mitochondria accumulates gangliosides. It could determine the artefactual MTT assay values at high GM1 concentrations, seen as lost in viability.

Figure 5a, b and c: In the representative traces, what do the bars represents?

Round 2

Reviewer 1 Report

The answer to my comments are fine except for the Wb figure3b and 4c. Your reply was: "we reported the best of three different experiments, the one with the highest resolution."

But how can you make a mean of fold increase if each immunostaining of your 3 experiments cannot be not shown due to poor resolution? what is the confidence in the mean data in that case?

You should be able to provide an original figure for 3b and a different one for 4c. Otherwise show only the one of figure 4c.

Author Response

Details on the revision according to the Editor and Reviewers’ comments and suggestions.

Comments

Point 1: The answer to my comments are fine except for the Wb figure3b and 4c. Your reply was: "we reported the best of three different experiments, the one with the highest resolution."

But how can you make a mean of fold increase if each immunostaining of your 3 experiments cannot be not shown due to poor resolution? what is the confidence in the mean data in that case?

Response 1: The resolution of the immunostaining was enough to obtain a good quantification, which is relative to a control and not quantitative. Indeed, we cannot report in the graph a specific amount of GM1 but the fold increase with respect to control. We decided to use in the paper the best one also in term of sample loading order to facilitate the comprehension of the figure.

Point 2: You should be able to provide an original figure for 3b and a different one for 4c. Otherwise show only the one of figure 4c.

Response 2: We change the figure 3b with another immunostaining.